# Neural Code Comprehension: A Learnable Representation of Code Semantics

**Tal Ben-Nun**
ETH Zurich
Zurich 8092, Switzerland
`talbn@inf.ethz.ch`

**Alice Shoshana Jakobovits**
ETH Zurich
Zurich 8092, Switzerland
`alicej@student.ethz.ch`

**Torsten Hoefler**
ETH Zurich
Zurich 8092, Switzerland
`htor@inf.ethz.ch`

## Abstract

With the recent success of embeddings in natural language processing, research has been conducted into applying similar methods to code analysis. Most works attempt to process the code directly or use a syntactic tree representation, treating it like sentences written in a natural language. However, none of the existing methods are sufficient to comprehend program semantics robustly, due to structural features such as function calls, branching, and interchangeable order of statements. In this paper, we propose a novel processing technique to learn code semantics, and apply it to a variety of program analysis tasks. In particular, we stipulate that a robust distributional hypothesis of code applies to both human- and machine-generated programs. Following this hypothesis, we define an embedding space, inst2vec, based on an Intermediate Representation (IR) of the code that is independent of the source programming language. We provide a novel definition of contextual flow for this IR, leveraging both the underlying data- and control-flow of the program. We then analyze the embeddings qualitatively using analogies and clustering, and evaluate the learned representation on three different high-level tasks. We show that even without fine-tuning, a single RNN architecture and fixed inst2vec embeddings outperform specialized approaches for performance prediction (compute device mapping, optimal thread coarsening); and algorithm classification from raw code (104 classes), where we set a new state-of-the-art.

## 1 Introduction

The emergence of the "Big Data era" manifests in the form of a dramatic increase in accessible code. In the year 2017 alone, GitHub reports [25] approximately 1 billion git commits (code modification uploads) written in 337 different programming languages. Sifting through, categorizing, and understanding code thus becomes an essential task for a variety of fields. Applications include identifying code duplication, performance prediction, algorithm detection for alternative code suggestion (guided programming), vulnerability analysis, and malicious code detection. These tasks are challenging, as code can be modified such that it syntactically differs (for instance, via different or reordered operations, or written in a different language altogether), but remains semantically equivalent (i.e., produces the same result). However, these tasks are also ideal for machine learning, since they can be represented as classic regression and classification problems.

In order to mechanize code comprehension, the research community typically employs reinforcement learning and stochastic compilation for *super-optimization* [13, 56]; or borrows concepts from Natural Language Processing (NLP) for human-authored code, relying on the following hypothesis:

> **The naturalness hypothesis [3].** *Software is a form of human communication; software corpora have similar statistical properties to natural language corpora; and these properties can be exploited to build better software engineering tools.*

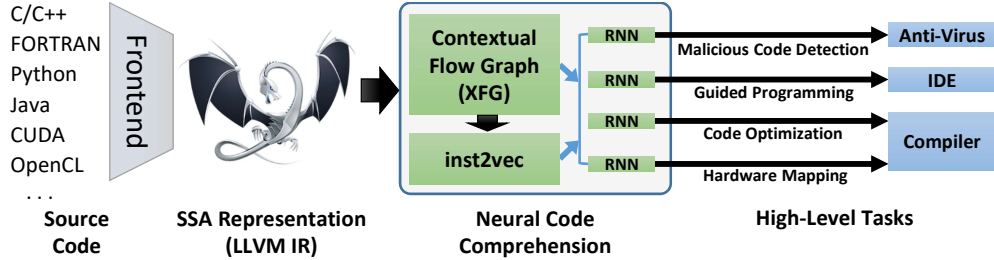

Figure 1: Component overview of the Neural Code Comprehension pipeline.

For NLP-based approaches, input code is usually processed into tokens (e.g., keywords, braces) [18] or other representations [4, 7, 53], and optionally undergoes embedding in a continuous lower-dimensional space. In the spirit of the successful `word2vec` model [47, 48], the mapping to the embedding space is learned by pairing a token with its surrounding tokens. Following this process, RNNs [23] are trained on sequences of such tokens. This model has been successfully used for NLP-like tasks, such as summarization [6], function name prediction [7], and algorithm classification [49].

Although the results for stochastic code optimization and NLP embeddings are promising, two issues arise. Firstly, in prior works, the source programming language (or machine code for optimization) is fixed, which does not reflect the plethora of languages, nor generalizes to future languages. Secondly, existing methods process tokens (or instructions) sequentially, targeting function- and loop-free code. Such codes, however, do not represent the majority of the applications.

This paper presents Neural Code Comprehension[1]: a general-purpose processing pipeline geared towards representing code semantics in a robust and learnable manner. The pipeline, depicted in Fig. 1, accepts code in various source languages and converts it to statements in an Intermediate Representation (IR), using the LLVM Compiler Infrastructure [39]. The LLVM IR, which is explained in detail in Section 4, is then processed to a robust representation that we call *conteXtual Flow Graphs (XFGs)*. XFGs are constructed from both the data- and control-flow of the code, thus inherently supporting loops and function calls. In turn, the XFG structure is used to train an embedding space for individual statements, called `inst2vec` (from the word "instruction"), which is fed to RNNs for a variety of high-level tasks.

Neural Code Comprehension is evaluated on multiple levels, using clustering and analogies for `inst2vec`, as well as three different code comprehension tasks for XFGs: algorithm classification; heterogeneous compute device (e.g., CPU, GPU) mapping; and optimal thread coarsening factor prediction, which model the runtime of an application without running it. Our datasets contain CPU and GPU code written in C, C++, OpenCL, and FORTRAN, though LLVM supports additional languages such as Python, Rust, Swift, Go, and CUDA. Our work makes the following contributions:

- We formulate a robust distributional hypothesis for code, from which we draw a novel distributed representation of code statements based on contextual flow and LLVM IR.
- We detail the construction of the XFG, *the first representation designed specifically for statement embeddings that combines data and control flow*.
- We evaluate the representation using clustering, analogies, semantic tests, and three fundamentally different high-level code learning tasks.
- Using one simple LSTM architecture and fixed pre-trained embeddings, we match or surpass the best-performing approaches in each task, including specialized DNN architectures.

## 2 Related Work

Distributed representations of code were first suggested by Allamanis et al. [2], followed by several works leveraging embeddings to apply NLP techniques to programming languages [3, 61].

**Code Representation** Previous research focuses on embedding high-level programming languages such as Java [20, 30], C [41], or OpenCL [18] in the form of *tokens* or statements, as well as lower

level representations such as object code [41]. To the best of our knowledge, however, no attempt has been made to train embeddings for compiler IRs prior to this work. As for representing the context of a token, which is necessary for training embeddings, some works rely on lexicographical locality [2, 18, 20], whereas others exploit the structural nature of code, using Data Flow Graphs [4], Control Flow Graphs [51, 53, 64], Abstract Syntax Trees (ASTs) [12, 30], paths in the AST [8], or an augmented AST, for instance with additional edges connecting different uses and updates of syntax tokens corresponding to variables [5]. We differ from all previous approaches by introducing contextual flow, a graph representation that captures both data and control dependencies. In compiler research, similar graphs exist but have not been successfully exploited for machine learning. Examples include the Program Dependence Graph (PDG) [24] and the IR known as Sea of Nodes [15, 16]. Unlike these representations, our graphs are not designed to be optimized by a compiler nor translated to machine code, which allows us to introduce ambiguity (e.g., ignoring parameter order) in favor of preserving context. Other works applying Machine Learning techniques to PDGs exist: Hsiao et al. [34] use PDGs to compute n-gram models for program analysis, and Wang et al. [62] use them for detecting copy direction among programs using Extreme Learning Machines. However, our work is the first to leverage a hybrid of control and data flow for the training of embeddings.

**Automated Tasks on Code**   Learned representations of code are commonly used for two types of tasks: uncovering program semantics or optimizing programs. For the former task, code embeddings have been used to perform function or variable naming [2, 7], clone detection [63], code completion [54, 65], summarization [6], and algorithm classification [49]. As for program optimization, research has been conducted on automatic feature generation for code [40, 50]; and Cummins et al. [18] notably leverage embeddings of OpenCL code to predict optimal device mapping and thread coarsening factors. Their work differs from ours in that the method is restricted to the OpenCL language, and that they process programs in a sequential order, which does not capture complex code structures. Furthermore, the state-of-the-art in automatic tuning for program optimization [10] uses surrogate performance models and active learning, and does not take code semantics into account.

**Embedding Evaluation**   Previous works that use code embeddings do not evaluate the quality of the trained space on its own merit, but rather through the performance of subsequent (downstream) tasks. One exception is Allamanis et al. [2], who present empirical evidence of vector similarities for similar method names. To the best of our knowledge, we are the first to quantify the quality of a code embedding space itself in the form of clustering, syntactic analogies, semantic analogies, and categorical distance tests.

## 3   A Robust Distributional Hypothesis of Code

The linguistic Distributional Hypothesis [32, 52] is given by: *Words that occur in the same contexts tend to have similar meanings*. We stipulate that code, which describes a sequence of operations to a processor, behaves similarly, and paraphrase this hypothesis to:

> ***Statements** that occur in the same **contexts** tend to have **similar semantics***.

However, the above wording is vague, due to the possible meanings of the highlighted elements. Below we attempt to provide adequate definitions, upon which we build a learnable code representation.

**Statements**   To choose the right abstraction for statements, we take two concerns into account: universality and uniformity. As stated above, source code comes in many languages and thus fixating on a single one would hinder universality. At the other extreme, machine code (assembly) is target-specific, containing specialized instructions and relying on hardware characteristics, such as registers and memory architectures. As for uniformity, in a high-level language one statement may represent simple arithmetics, multiple operations, or even class definitions (for example, the Java statement `button.setOnClickListener(new View.OnClickListener(){...})`). On the other hand, assembly is too limited, since instructions are reused for different purposes. We thus wish to choose statements that are independent of the source language, as well as the hardware architecture.

**Context**   The definition of a context for code statements should also be carefully considered. We define context as *statements whose execution directly depends on each other*. Learning from consecutive statements in code does not necessarily fulfill this definition, as, for example, a programmer

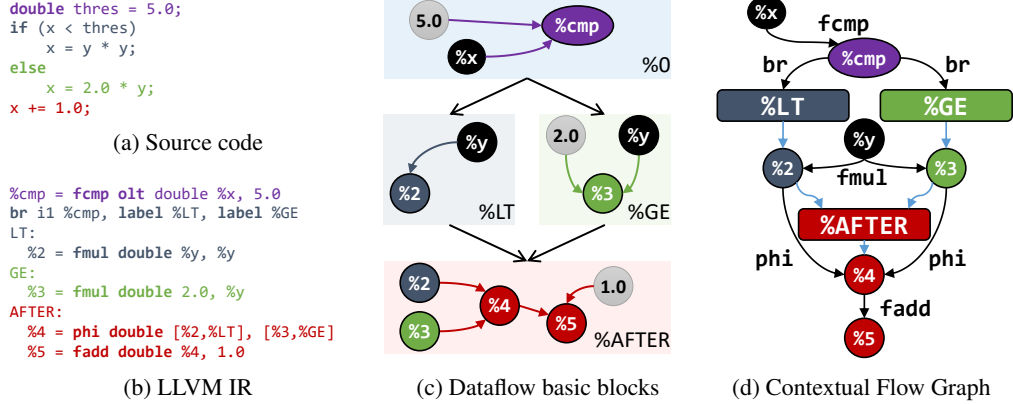

(a) Source code

(b) LLVM IR    (c) Dataflow basic blocks    (d) Contextual Flow Graph

Figure 2: Contextual flow processing scheme.

may use a variable in the first line of a function, but only use it again in the last line. Moreover, such long-term relationships may vanish when using RNNs and attention learning. It is possible to determine the data dependencies of each statement by analyzing dataflow, however, branches and function calls do not necessarily generate such dependencies. Another way of representing execution dependence is through the notion of causality (i.e., the "happens-before" relation [38]), which can be used to complement dataflow. In our representation, context is the union of data dependence and execution dependence, thereby capturing both relations.

**Similarity**   To define similarity, one first needs to define the *semantics* of a statement. We draw the definition of semantics from Operational Semantics in programming language theory, which refers to the effects (e.g., preconditions, postconditions) of each computational step in a given program. In this paper, we specifically assume that each statement modifies the system state in a certain way (e.g., adds two numbers) and consumes resources (e.g., uses registers and floating-point units). It follows that semantic similarity can be defined by two statements consuming the same resources or modifying the system state in a similar way. Using this definition, two versions of the same algorithm with different variable types would be synonymous.

## 4   Contextual Flow Processing

The aforementioned statements and contexts cannot be directly extracted from source code, but rather require processing akin to partial compilation (e.g., dataflow extraction). In this section, we briefly describe a popular compilation pipeline and proposed modifications to create a learnable vocabulary of statements and their context.

### 4.1   Compilation, Static Single Assignment, and LLVM IR

Major contemporary compilers, such as GCC and LLVM, support multiple programming languages and hardware targets. To avoid duplication in code optimization techniques, they enforce a strict separation between the source language (frontend), an Intermediate Representation (IR) that can be optimized, and the target machine code (backend) that should be mapped to a specific hardware. In particular, the LLVM IR [45] supports various architectures (e.g., GPUs), and can represent optimized code (e.g., using vector registers) inherently. Figures 2a and 2b depict an example code and its LLVM IR equivalent, and the structure of an LLVM IR statement is shown in Fig. 3.

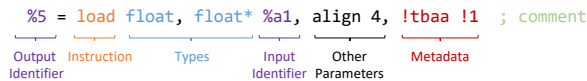

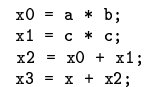

Figure 3: Anatomy of an LLVM IR statement.    Figure 4: SSA of `x += (a*b)+(c*c)`.

In the LLVM infrastructure, the IR is given in Static Single Assignment (SSA) form [19]. Briefly, an SSA IR ensures that every variable is assigned only once, which makes it easy to track dataflow between IR statements, as shown in Fig. 4. To overcome analysis issues resulting from control-flow, such as loops, SSA defines $\phi$-expressions. These expressions enumerate all possible outcomes that

can lead to a variable (depending on the runtime control-flow), and can be used to optimize code across branches. In Fig. 2b, the identifier `%4` is constructed from a $\phi$-expression that can take either the value of `%2` or `%3`, depending on the value of `x`.

## 4.2 Contextual Flow Graphs

To analyze dataflow for optimization, LLVM divides the IR statements into "basic blocks", which contain no control-flow divergence, illustrated in Fig. 2c. Within a basic block, statements naturally create traceable dataflow as SSA lists data dependencies in the form of input identifiers (even if conditional), and assigns the results to a single identifier. However, as shown in Section 3, dataflow alone does not suffice to provide context for a given statement, e.g., when in the vicinity of a branch. Therefore, we define a representation that incorporates both the relative data- and control-flow of a statement, which we call the **conteXtual Flow Graph (XFG)**.

XFGs (e.g., Fig. 2d) are directed multigraphs, where two nodes can be connected by more than one edge. XFG nodes can either be variables or label identifiers (e.g., basic block, function name), appearing in the figure as ovals or rectangles respectively. Correspondingly, an edge either represents data-dependence (in black), carrying an LLVM IR statement; or execution dependence (light blue).

**XFG Construction**   We generate XFGs incrementally from LLVM IR, as follows:

1. Read LLVM IR statements once, storing function names and return statements.

2. Second pass over the statements, adding nodes and edges according to the following rule-set:
   (a) Data dependencies within a basic block are connected.
   (b) Inter-block dependencies (e.g., $\phi$-expressions) are both connected directly and through the label identifier (statement-less edges).
   (c) Identifiers without a dataflow parent are connected to their root (label or program root).

It follows that XFGs create paths through dataflow as well as branches, loops, and functions (including recursion). Owing to the two passes, as well as the linear-time construction of LLVM IR [58], XFGs are constructed in $\mathcal{O}(n)$ for a program with $n$ SSA statements. This is especially valuable when learning over large code corpora, such as Tensorflow.

**External Code**   Calls to external code (e.g., libraries, frameworks) can be divided into two categories: statically- and dynamically-linked. If the code is accessible during compilation (header-only frameworks and static libraries), LLVM IR is available and the statements are traversed as part of the XFG. In the dynamic case, the library code is not included and is represented as a `call` statement.

## 5   `inst2vec`: Embedding Statements in Continuous Space

With XFGs providing a notion of context, we can now train an embedding space for individual statements. To support learnability, desiderata for such a space include: (a) statements that are in close proximity should have similar artifacts on a system (i.e., use the same resources); and (b) changing the same attributes (e.g., data type) for different instructions should result in a similar offset in the space. We train LLVM IR statement embeddings using the skip-gram model [48], following preprocessing to limit the vocabulary size.

### 5.1   Statement Preprocessing and Training

**Preprocessing**   First, we filter out comments and metadata from statements. Then, identifiers and immediate values (numeric constants, strings) are replaced with `%ID` and `<INT/FLOAT/STRING>` respectively, where immediate values are fed separately to downstream RNNs. Lastly, data structures are "inlined", that is, their contents are encoded within the statement. Fig. 5 lists statements before and after preprocessing.

```
store float %250, float* %82, align 4, !tbaa !1      store float %ID, float* %ID, align 4
%10 = fadd fast float %9, 1.3                         %ID = fadd fast float %ID, <FLOAT>
%8 = load %"struct.aaa"*, %"struct.aaa"** %2          %ID = load { float, float }*, { float, float }** %ID
```

<div align="center">

(a) LLVM IR                                     (b) `inst2vec` statements

Figure 5: Before and after preprocessing LLVM IR to `inst2vec` statements.

</div>

Table 1: `inst2vec` training dataset statistics

| Discipline | Dataset | Files | LLVM IR Lines | Vocabulary Size | XFG Stmt. Pairs |
|---|---|---|---|---|---|
| Machine Learning | Tensorflow [1] | 2,492 | 16,943,893 | 220,554 | 260,250,973 |
| High-Performance Computing | AMD APP SDK [9] | 123 | 1,304,669 | 4,146 | 45,081,359 |
| | BLAS [22] | 300 | 280,782 | 566 | 283,856 |
| Benchmarks | NAS [57] | 268 | 572,521 | 1,793 | 1,701,968 |
| | Parboil [59] | 151 | 118,575 | 2,175 | 151,916 |
| | PolybenchGPU [27] | 40 | 33,601 | 577 | 40,975 |
| | Rodinia [14] | 92 | 103,296 | 3,861 | 266,354 |
| | SHOC [21] | 112 | 399,287 | 3,381 | 12,096,508 |
| Scientific Computing | COSMO [11] | 161 | 152,127 | 2,344 | 2,338,153 |
| Operating Systems | Linux kernel [42] | 1,988 | 2,544,245 | 136,545 | 5,271,179 |
| Computer Vision | OpenCV [36] | 442 | 1,908,683 | 39,920 | 10,313,451 |
| | NVIDIA samples [17] | 60 | 43,563 | 2,467 | 74,915 |
| Synthetic | Synthetic | 17,801 | 26,045,547 | 113,763 | 303,054,685 |
| Total (Combined) | — | 24,030 | 50,450,789 | 8,565 | 640,926,292 |

**Dataset**   Table 1 summarizes the code corpora and vocabulary statistics of the `inst2vec` dataset. We choose corpora from different disciplines, including high-performance computing, benchmarks, operating systems, climate sciences, computer vision, machine learning (using Tensorflow's own source code), and synthetically-generated programs. The code in the dataset is written in C, C++, FORTRAN, and OpenCL, and is compiled for Intel CPUs as well as NVIDIA and AMD GPUs. The files in the dataset were compiled to LLVM IR with Clang [44] and Flang [43], using compilation flags from the original code (if available) and randomly chosen compiler optimization (e.g., `-ffast-math`) and target architecture flags.

For the synthetic corpus, we use both C code and the Eigen [31] C++ library. In particular, random linear algebra operations are procedurally generated from high-level templates using different parameters, such as data types, operations, and dimensions.

**Setup and Training**   Given a set of XFGs created from the LLVM IR files, we generate neighboring statement pairs up to a certain context size, following the skip-gram model [48]. A context of size $N$ includes all statement pairs that are connected by a path shorter or equal to $N$. To obtain the pairs, we construct a dual graph in which statements are nodes, omitting duplicate edges. Following this process, we discard statements that occur less than 300 times in the dataset, pairs of identical statements, and perform subsampling of frequent pairs, similarly to Mikolov et al. [48]. We train `inst2vec` with an embedding dimension of 200 for 5 epochs using Tensorflow [1]. The Adam optimizer [37] is used with the default published hyperparameters and softmax cross-entropy loss.

## 5.2   Evaluation

**Clustering**   Fig. 6 depicts the t-SNE [60] plots for trained `inst2vec` spaces with different XFG context sizes, colored by statement and data type (legend in Appendix A). In the plots, we see that both a context size of 1 statement in each direction (Fig. 6a) or 3 statements (Fig. 6c) generate large, multi-type clusters, as well as outliers. This phenomenon eventually contributes to a lower final analogy score, due to inappropriate representation of inter-statement relations, as can be seen below. Owing to these results, we choose a context size of 2 statements (Fig. 6b), which mostly consists of separate, monochromatic clusters, indicating strong clustering w.r.t. instruction and data types. While data type syntactic clusters are unsurprising, their existence is not trivial, since the dataset contains diverse codebases rather than copies of the same functions with different types.

An example of a *semantically*-similar statement cluster can be found in data structures. In particular, the top-5 nearest neighbors of operations on the complex data type "`std::complex<float>`" include "`2 x float`" (i.e., a vector type). In fact, LLVM IR represents the complex data type as `{float, float}`, so this property is generalized to any user-defined data structure (`struct`) with two floats.

**Analogies and Tests**   We also evaluate `inst2vec` by automatically generating a list of statement analogies ("a" is to "b" as "c" is to "?", or "`a:b; c:?`") that appear in our vocabulary using the LLVM

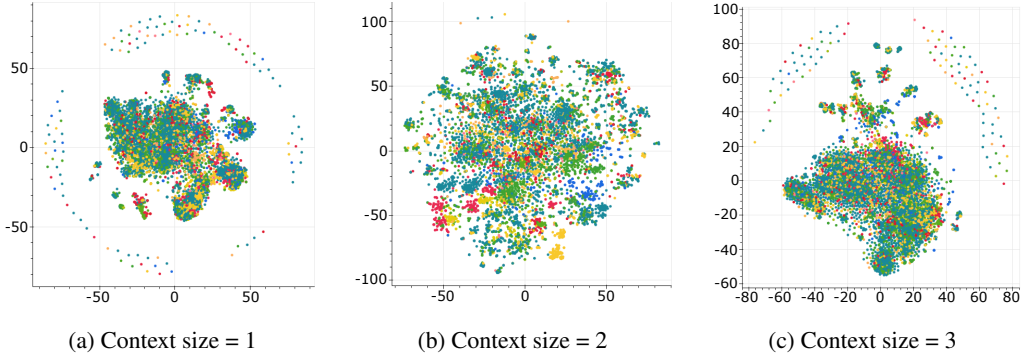

| (a) Context size = 1 | (b) Context size = 2 | (c) Context size = 3 |

Figure 6: Two-dimensional t-SNE plots for learned embeddings (best viewed in color).

IR syntax. We then use the embeddings to find the result by computing `a-b+c` and asking whether the result is in the top-5 neighbors (cosine distance). Additionally, we automatically create relative distance expressions using the LLVM IR reference categories [45] of the form $d(a, b) < d(a, c)$ to test whether statements that use different resources are further away than those who use the same.

Table 2 shows the analogy and test results for `inst2vec` trained on XFG as well as on CFG (control flow-only) and DFG (data flow-only) for different context sizes. The analogies are divided into different categories, including data types (i.e., transitions between types), options (e.g., fast math), conversions (e.g., bit casting, extension, truncation), and data structures (e.g., vector-type equivalents of structures). Below are examples of a type analogy:

```
%ID = add i64 %ID, %ID   :   %ID = fadd float %ID, %ID;
%ID = sub i64 %ID, %ID   :?  %ID = fsub float %ID, %ID
```

and a data structure analogy:

```
%ID = extractvalue { double, double } %ID, 0   :   %ID = extractelement <2 x double> %ID, <TYP> 0;
%ID = extractvalue { double, double } %ID, 1   :?  %ID = extractelement <2 x double> %ID, <TYP> 1
```

The results confirm that over all scores, a context size of 2 is the best-performing configuration, and show that the XFG representation is more complete and leads to better embeddings than taking into account control or data flow alone.

Table 2: Analogy and test scores for `inst2vec`

| Context type | Context Size | Syntactic Analogies | | Semantic Analogies | | Semantic Distance Test |
|---|---|---|---|---|---|---|
| | | Types | Options | Conversions | Data Structures | |
| CFG | 1 | 0 (0 %) | 1 (1.89 %) | 1 (0.07 %) | 0 (0 %) | 51.59 % |
| | 2 | 1 (0.18 %) | 1 (1.89 %) | 0 (0 %) | 0 (0 %) | 50.47 % |
| | 3 | 0 (0 %) | 1 (1.89 %) | 4 (0.27 %) | 0 (0 %) | 53.79 % |
| DFG | 1 | 53 (9.46 %) | 12 (22.64 %) | 2 (0.13 %) | 4 (50.00 %) | 56.79 % |
| | 2 | 71 (12.68 %) | 12 (22.64 %) | 12 (0.80 %) | 3 (37.50 %) | 57.44 % |
| | 3 | 67 (22.32 %) | 18 (33.96 %) | 40 (2.65 %) | 4 (50.00 %) | 60.38 % |
| XFG | 1 | 101 (18.04 %) | 13 (24.53 %) | 100 (6.63 %) | 3 (37.50 %) | 60.98 % |
| | 2 | **226 (40.36 %)** | **45 (84.91 %)** | **134 (8.89 %)** | **7 (87.50 %)** | **79.12 %** |
| | 3 | 125 (22.32 %) | 24 (45.28 %) | 48 (3.18 %) | **7 (87.50 %)** | 62.56 % |

# 6   Code Comprehension Experiments

In this section, we evaluate `inst2vec` on three different tasks, comparing with manually-extracted features and state-of-the-art specialized deep learning approaches. Throughout all tasks, we use *the same neural network architecture* and our pre-trained embedding matrix from Section 5, which remains fixed during training.

**Training**   Our recurrent network (see schematic description in the Appendix B) consists of an `inst2vec` input with an XFG context size of 2, followed by two stacked LSTM [33] layers with 200 units in each layer, batch normalization [35], a dense 32-neuron layer with ReLU activations, and output units matching the number of classes. The loss function is a categorical cross-entropy trained

using Adam [37] with the default hyperparameters. Additionally, for the compute device mapping and optimal thread coarsening factor prediction tasks, we train the LLVM IR statements with the immediate values that were stripped from them during preprocessing (see Section 5). Further details are given in Appendix C.

**Datasets** The algorithm classification task uses the POJ-104 [49] dataset[2], collected from a Pedagogical Open Judge system. The dataset contains 104 program classes written by 500 different people (randomly selected subset per class). For the compute device mapping and optimal thread coarsening factor prediction tasks, we use an OpenCL code dataset[3] provided by Cummins et al. [18].

## 6.1 Algorithm Classification

Using `inst2vec`, we construct an RNN that reads embedded source code and outputs a predicted program class. We compare our approach with Tree-Based CNNs (TBCNN) [49], the best-performing algorithm classifier in the POJ-104 dataset. TBCNN constructs embeddings from Astract Syntax Tree nodes of source code, and employs two specialized layers: tree convolutions and dynamic pooling. Their network comprises 5 layers, where convolution and fully connected layers are 600-dimensional. Our data preparation follows the experiment conducted by Mou et al. [49], splitting the dataset 3:1:1 for training, validation, and testing. To compile the programs successfully, we prepend `#include` statements to each file. Data augmentation is then applied on the training set by compiling each file 8 times with different flags (`-O{0-3}`, `-ffast-math`).

Table 3: Algorithm classification test accuracy

| Metric | Surface Features [49] (RBF SVM + Bag-of-Trees) | RNN [49] | TBCNN [49] | inst2vec |
|---|---|---|---|---|
| Test Accuracy [%] | 88.2 | 84.8 | 94.0 | **94.83** |

Table 3 compares `inst2vec` (trained for 100 epochs) with the reported results of Mou et al. [49], which contain TBCNN as well as a 600-cell RNN and a manual feature extraction approach (Surface Features). The results show that `inst2vec` sets a new state-of-the-art with a 13.8 % decrease in error, even though the dataset used to generate the embeddings *does not include POJ-104* (see Table 1).

## 6.2 Heterogeneous Compute Device Mapping

Next, we use Neural Code Comprehension to predict whether a given OpenCL program will run faster on a CPU (Intel Core i7-3820) or a GPU (AMD Tahiti 7970 and NVIDIA GTX 970) given its code, input data size, and *work-group size* (i.e., number of threads that work in a group with shared memory). To achieve that, we use the same experimental methodology presented by Cummins et al. [18], removing their specialized OpenCL source rewriter and replacing their code token embeddings with our XFGs and `inst2vec`. We concatenate the data and work-group sizes to the network inputs, and train with stratified 10-fold cross-validation. We repeat the training 10 times with random initialization of the network's weights and report the best result.

In Table 4, `inst2vec` and `inst2vec-imm` (i.e., with immediate value handling) are compared with a manual code feature extraction approach by Grewe et al. [29] and DeepTune [18], in terms of runtime prediction accuracies and resulting speedup. The baseline for the speedup is a static mapping, which selects the device that yields the best average case performance over all programs in the data set: in the case of AMD Tahiti versus Intel i7-3820, that is the CPU and in the case of NVIDIA GTX versus Intel i7-3820, it is the GPU. The results indicate that `inst2vec` outperforms Grewe et al. and is on-par with DeepTune. We believe that the better predictions in DeepTune are the result of training the embedding matrix in tandem with the high-level task, thereby specializing it to the dataset. This specialized training is, however, surpassed by taking immediate values into account during training. We present the result of the best immediate value handling method in Table 4 (`inst2vec-imm`), and the exhaustive results can be found in Appendix D.

Table 4: Heterogeneous device mapping results

| Architecture | Prediction Accuracy [%] | | | | |
|---|---|---|---|---|---|
| | GPU | Grewe et al. [29] | DeepTune [18] | inst2vec | inst2vec-imm |
| AMD Tahiti 7970 | 41.18 | 73.38 | 83.68 | 82.79 | **88.09** |
| NVIDIA GTX 970 | 56.91 | 72.94 | 80.29 | 82.06 | **86.62** |
| | Speedup | | | | |
| | GPU | Grewe et al. | DeepTune | inst2vec | inst2vec-imm |
| AMD Tahiti 7970 | 3.26 | 2.91 | 3.34 | 3.42 | **3.47** |
| NVIDIA GTX 970 | 1.00 | 1.26 | 1.41 | 1.42 | **1.44** |

## 6.3 Optimal Thread Coarsening Factor Prediction

Our third example predicts the best-performing *thread coarsening factor*, a measure of the amount of work done per GPU thread, for a given OpenCL code. We again compare the achieved speedups of `inst2vec` with manual features [46], DeepTune, and DeepTune with transfer learning applied from the task in Section 6.2 (denoted by DeepTune-TL). Possible values for the coarsening factor are 1 (baseline for speedups), 2, 4, 8, 16, and 32. The results in Table 5 show that while `inst2vec` yields better speedups than DeepTune-TL in only half of the cases (possibly due to the embedding specialization in DeepTune), the manually-extracted features are consistently outperformed by `inst2vec`. Moreover, `inst2vec-imm` is consistently on-par with DeepTune, but improves inconsistently on `inst2vec` (on the AMD Tahiti and the NVIDIA GTX only), and fails to outperform DeepTune-TL. This can be explained by the small size of the training data for this task (17 programs with 6 different thread coarsening factors for each hardware platform). The optimal device mapping task (Section 6.2), on the other hand, features 680 programs for each platform.

Table 5: Speedups achieved by coarsening threads

| Computing Platform | Magni et al. [46] | DeepTune [18] | DeepTune-TL [18] | inst2vec | inst2vec-imm |
|---|---|---|---|---|---|
| AMD Radeon HD 5900 | 1.21 | 1.10 | 1.17 | **1.37** | 1.28 |
| AMD Tahiti 7970 | 1.01 | 1.05 | **1.23** | 1.10 | 1.18 |
| NVIDIA GTX 480 | 0.86 | 1.10 | **1.14** | 1.07 | 1.11 |
| NVIDIA Tesla K20c | 0.94 | 0.99 | 0.93 | **1.06** | 1.00 |

# 7 Conclusion

In this paper, we have empirically shown that semantics of statements can be successfully recovered from their context alone. This recovery relies both on proper granularity, where we propose to use filtered LLVM IR instructions; and on the grouping of statements, for which we use a mixture of data- and control-flow. We use our proposed representation to perform three high-level classification and prediction tasks, outperforming all manually-extracted features and achieving results that are on-par with (and better than) two inherently different state-of-the-art specialized DNN solutions.

With this work, we attempt to pave the way towards mechanized code comprehension via machine learning, whether the code was authored by a human or automatically-generated. Further research could be conducted in various directions. Rather than directly using statements, the representation may be refined using part-based models, which have already been applied successfully in language models [55]. `inst2vec` can also be used as a basis for neural code interpretation, using a modified Differentiable Neural Computer [28] to enable execution of arbitrary code over DNNs.

**Acknowledgments**

We wish to thank Theodoros Theodoridis, Kfir Levy, Tobias Grosser, and Yunyan Guo for fruitful discussions. The authors also acknowledge MeteoSwiss, and thank Hussein Harake, Colin McMurtrie, and the whole CSCS team for granting access to the Greina machines, and for their excellent technical support. TBN is supported by the ETH Postdoctoral Fellowship and Marie Curie Actions for People COFUND program.

## Footnotes

[1]Code, datasets, trained embeddings, and results available at `https://www.github.com/spcl/ncc`

[2]`https://sites.google.com/site/treebasedcnn/`

[3]`https://www.github.com/ChrisCummins/paper-end2end-dl`

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
