[Supplementary Material · supplementary.pdf]

# Neural Code Comprehension: A Learnable Representation of Code Semantics — Supplementary Information

**Tal Ben-Nun**
ETH Zurich
Zurich 8092, Switzerland
`talbn@inf.ethz.ch`

**Alice Shoshana Jakobovits**
ETH Zurich
Zurich 8092, Switzerland
`alicej@student.ethz.ch`

**Torsten Hoefler**
ETH Zurich
Zurich 8092, Switzerland
`htor@inf.ethz.ch`

## Abstract

This file contains the appendices of the paper "Neural Code Comprehension: A Learnable Representation of Code Semantics". The code, datasets, trained embeddings, and results are available at `https://www.github.com/spcl/ncc`

# A  Statement Categories for `inst2vec` Clustering Results

Table 1 presents the mapping from colors to statement categories that appear in Fig. 6. The following rules apply to the categories in the table:

1. A `type operation` generally refers to an operation, a function call, or the definition of a function, that returns an instance of `type`.

2. `type*` refers to a pointer of `type`. Asterisks could be chained for pointers-to-pointers.

3. `<d x type>` is a vector of `d` elements of `type`.

4. `[d x type]` is an array of `d` elements of `type`.

5. `struct/class` denotes an aggregate structure (e.g., C `struct`) of multiple types, e.g., `{type_1, type_2, ..., type_n}` in LLVM IR.

6. `floating point` can refer to either single- or double-precision floating point values.

7. `int` can refer to an integer of any bit-width.

8. `void` categories (`call void`, `invoke void`) refer to calls/invocations of functions that have no return value.

9. `conversion operations` denote type conversions within LLVM, which do not necessarily translate into code.

10. `load function pointer`, `store function pointer` refer to instructions that read or write function pointers into memory, respectively.

Table 1: Statement category by color (Fig. 6 legend)

| Color | Statement Category | Example |
|---|---|---|
| ● | `<d x int>* operation` | `<%ID> = load <2 x i64>*, <2 x i64>** <%ID>, align 8` |
| ● | `<d x int> operation` | `<%ID> = and <8 x i32> <%ID>, <%ID>` |
| ● | `<d x struct/class*> operation` | `store <2 x { i64, i64 }*> <%ID>, <2 x { i64, i64 }*>* <%ID>, align 8` |
| ● | `struct/class* operation` | `<%ID> = phi { float, float }* [ <%ID>, <%ID> ], [ <%ID>, <%ID> ]` |
| ● | `struct/class operation` | `<%ID> = alloca { i32, i32 }, align 4` |
| ● | `int** operation` | `<%ID> = phi i8** [ <%ID>, <%ID> ], [ <%ID>, <%ID> ]` |
| ● | `int* operation` | `<%ID> = load i8*, i8** <%ID>, align 8` |
| ● | `int operation` | `<%ID> = add i16 <%ID>, <INT>` |
| ● | `type conversion operation` | `<%ID> = bitcast <4 x i32> <%ID> to <16 x i8>` |
| ● | `global variable definition` | `<@ID> = global i32 <INT>, align 4` |
| ● | `<d x int*> operation` | `<%ID> = phi <4 x i8*> [ <%ID>, <%ID> ], [ <%ID>, <%ID> ]` |
| ● | `load function pointer` | `<%ID> = load { i32 (...)** }*, { i32 (...)** }** <%ID>, align 8` |
| ● | `store function pointer` | `store void ()* <@ID>, void ()** <%ID>, align 8` |
| ● | `floating point** operation` | `<%ID> = phi float** [ <%ID>, <%ID> ], [ <%ID>, <%ID> ]` |
| ● | `floating point* operation` | `<%ID> = icmp eq double* <%ID>, null` |
| ● | `floating point operation` | `<%ID> = getelementptr double, double* <%ID>, i64 <%ID>` |
| ● | `call void` | `tail call void <@ID>(i64 <INT>)` |
| ● | `other/misc.` | `cleanup; unreachable` |
| ● | `[d x [d x type]] operation` | `<%ID> = getelementptr inbounds [8 x [256 x i32]], [8 x [256 x i32]]*` |
| ● | `[d x struct/class] operation` | `<%ID> = alloca [5 x { i8*, i64 }], align 8` |
| ● | `[d x int] operation` | `<%ID> = alloca [100 x i8], align 16` |
| ● | `[d x floating point] operation` | `<%ID> = getelementptr inbounds [1024 x double], [1024 x double]*` |
| ● | `<d x floating point>* operation` | `<%ID> = alloca <8 x float>*, align 8` |
| ● | `<d x floating point> operation` | `<%ID> = call <4 x float> <@ID>(float* <%ID>)` |
| ● | `void function definition` | `define linkonce_odr void <@ID>({ i32 (...)** }*) unnamed_addr` |
| ● | `invoke void` | `invoke void <@ID>(i8* <%ID>) to label <%ID> unwind label <%ID>` |

# B    Neural Code Comprehension: Network Architecture

Fig. 1 depicts the neural network architecture used for the high-level tasks in this paper. Below we describe each of the underlying layers in the network.

**Input and Embedding Lookup**   As an input, the Neural Code Comprehension architecture accepts programs as sequences of LLVM IR statements. Each statement is represented through its corresponding embedding vector, and for statements that are not in the `inst2vec` vocabulary, they are assigned the embedding vector corresponding to a predefined "unknown" token. The embedding layer remains fixed throughout the training of the code comprehension tasks (effectively, it acts as a simple lookup matrix), and no fine-tuning is applied to the vector representations.

**Program Characterization**   The sequence of statement embedding vectors is passed to two layers of Long Short-Term Memory (LSTM) [33] cells. This program characterization layer transforms an input sequence of arbitrary length into a fixed-length vector that captures the properties of the processed program.

**Auxiliary Input Concatenation (optional)**   Additional data may optionally be concatenated with the output of the two-layer LSTM at this point. This allows information that is only available at runtime (e.g., hardware parameters or data size) to be taken into account in the predictive modeling.

**Batch normalization** [35] is performed, and then the vector output of program characterization goes through a 32-unit fully connected **dense** layer with rectifier (ReLU) activations [26]. Finally, the **output** layer is another fully-connected layer, which features a number of units equal to the number of possible output categories. The output is given by a sigmoid activation function (output between $0$ and $1$), where the largest activation corresponds to the model's prediction.

Figure 1: Neural Code Comprehension Network Architecture

# C    Training NCC with Immediate Values: Method Description

In the transformations applied to raw LLVM IR code before `inst2vec` training, the statements are stripped of their immediate values and are replaced by tokens indicating the value type: `<INT>`, `<FLOAT>`, `<STRING>` (see Section 5 for further detail). The purpose of this abstraction from the immediate values is twofold. First, keeping all immediate values would result in an extremely large and sparse vocabulary size; second, this transformation allows to map statements with nearly identical semantics (e.g. `<%ID> = fadd fast float <%ID>, 1.3` and `<%ID> = fadd fast float <%ID>, 5.2`) to the same embedding vector. While this choice is sound for statement training, it might nevertheless fall short in the training of downstream tasks, where immediate values may hold values critical to the program's semantics or performance, such as array sizes or iteration bounds. In order to train `inst2vec` program representations along with their immediate values, we store the immediate values of each statement separately before filtering them out during preprocessing. The immediate values are then reintegrated into the NCC workflow using one of the three methods illustrated in Fig. 2 and described below.

(a) `concat_naïve`

(b) `concat_embed`

(c) `extract_concat`

Figure 2: Three architectures for training `inst2vec` sequences of statements along with their immediate values in NCC. The components related to immediate values are marked in dark orange. The stage at which the immediate values are concatenated with the statements is denoted with a yellow "+" sign.

**"naïve concatenation"** (`concat_naïve`)  Instead of feeding the model with the embedding vector of a statement alone (see layer 2, above), embedding vectors are first concatenated with their corresponding immediate values. The first set of LSTM cells accept an input of size *embedding dimension + length of list of immediates*. The remainder of the NCC layers are unchanged.

**"concatenate then embed"** (`concat_embed`)  This method introduces an additional embedding step: statement embedding vectors are first concatenated with their corresponding immediate values. They then pass through a fully-connected layer, which reduces the layer dimension from *input dimension = embedding dimension + length of list of immediates* back to *embedding dimension*. Next, program characterization and the rest of the network remain unchanged.

**"extract then concatenate"** (`extract_concat`)  In this method, immediate values are never coupled back directly to the statement from which they were extracted. Rather, the sequence of immediate values of the entire program undergoes a separate processing pipeline, before being concatenated with the output of the program characterization as auxiliary inputs (see Fig. 1). The separate processing of immediate values sequence consists of a single LSTM layer, designed to extract the critical information from the immediate values that characterize the program.

## D  Training NCC with Immediate Values: Exhaustive Results

Tables 2 and 3 present the results for the heterogeneous device mapping and optimal thread coarsening factor tasks, obtained with the different modes of immediate value handling described in Appendix C. The column 'ignore' presents the results for the simplest version of NCC, where immediate values are ignored.

Table 2: Heterogeneous device mapping results obtained with `inst2vec` and NCC, using different modes of immediate value handling

| Architecture | Prediction Accuracy [%] | | | | Speedup | | | |
|---|---|---|---|---|---|---|---|---|
| | ignore | concat naïve | extract concat | concat emb | ignore | concat naïve | extract concat | concat emb |
| AMD Tahiti 7970 | 82.79 | **88.09** | 76.18 | 72.06 | 3.42 | **3.47** | 3.36 | 2.76 |
| NVIDIA GTX 970 | 82.06 | **86.62** | 79.71 | 72.50 | 1.42 | **1.44** | 1.40 | 1.32 |

Table 3: Speedups achieved by coarsening threads with `inst2vec` and NCC, using different modes of immediate value handling

| Computing Platform | Speedup | | | |
|---|---|---|---|---|
| | ignore | concat_naïve | extract_concat | concat_embed |
| AMD Radeon HD 5900 | **1.37** | 1.21 | 1.28 | 1.30 |
| AMD Tahiti 7970 | 1.10 | 1.06 | **1.18** | 0.92 |
| NVIDIA GTX 480 | 1.07 | 0.99 | **1.11** | 0.97 |
| NVIDIA Tesla K20c | **1.06** | 1.04 | 1.00 | 0.99 |