[Reviews · NeurIPS 2018]

Reviewer 1



The paper presented a way of learning statement embeddings and demonstrated its effectiveness through empirical studies (from analogy tests and as inputs to downstream tasks around code comprehension). The paper is largely easy to follow and well written, although I'd still like to see improvements in the following aspects - the main technical novelty here seems to be defining the proper "context" and "granularity" for statements to then allow the skip-gram model to be naturally applied. I would therefore like to see some comparisons between the different configurations (even all under the same framework proposed in the paper, so sth. like an ablation study), e.g. when only data or control flow is considered, to gain more insights/justifications for the adopted paradigm. - in L.200 on P.6, please elaborate what you mean exactly by "colored by statement and data type". - for the 3 code comprehension experiments, please clarify whether the embedding layer, once initialized from inst2vec embeddings, is kept fixed during training, or still being fine-tuned along with the other network parameters.

Reviewer 2



This paper proposes to use programming code embeddings for individual statements to be used in all kinds of program analysis tasks, such as classification of type of algorithms, placing of code on a heterogeneous cpu-gpu architecture and scheduling of programs on gpus. The novelty of the paper consists of the way the code embeddings are computed/trained. Instead of looking at code statements in high-level languages and following either data or control flow, the paper proposes to use statements at an intermediate-representation level (which are independent of the high-level language used) and take both data and control flow into account. As such, the proposed technique builds "contextual flow graphs" where the nodes are connected either through data or control flow edges. The nodes are variable or label identifiers. The statements embeddings are computed using the skip-gram model on paths of the contextual flow graphs. The embeddings are trained on a sizeable dataset that includes the code of tensorflow, gpu benchmarks, linux kernel and synthetic code. The trained embeddings are used for three different predictive tasks: classification of algorithm type, placing code on a cpu or a gpu, and optimizations for gpu scheduling. In general, the embeddings provide competitive results when compared to either hand-tuned versions or state of the art baselines. I think this paper is a nice bridge from traditional program analysis and machine learning. For a researcher with experience in program analysis the contextual graphs are somewhat intuitive. The authors choose to represent variables as nodes instead of whole sentences. It would be nice to provide some intuition why that performs better or is a better choice than having nodes as statements (while keeping the edges to represent both data and control flow). I would have liked to see some simple experiments with code similarity. One idea for such an experiment: take two implementations for the same algorithm and try to show that in terms of code embeddings they appear similar (it's common to find the implementation of the same neural network in two different frameworks, such as tensorflow and pytorch). The authors could include a discussion on the limitation of the analysis. In particular, a discussion about how library code is treated. For the suggestion above with two different frameworks, would the embeddings work? Or would the information be lost due to library calls and the reliance of frameworks? For which type of tasks are code embeddings expected to work and for which ones they don't provide sufficient information? I'm not sure "semantics" is the right term to use for characterizing similarity wrt to consuming/producing similar resources. In general, "semantics" makes me think of a deeper meaning (potentially expressed at a higher granularity than one IR statement).

Reviewer 3



This paper discusses learning a general embedding representation to capture semantics in code, which can be used for multiple tasks. This is very important topic in the field right now, as there is a growing interest in applying learning approaches to program analysis tasks. The lack of general embedding requires to train a separate embedding network per task; however, the lack of available labeled data makes this impractical. Therefore, the goal of this work can mitigate this issue, and accelerates the adoption of deep learning approaches in the classical program analysis world. While the goal is great, my main concerns about this paper are two-folds. First, the proposed techniques are a bit shallow. This paper mainly proposes a conteXtual Flow Graphs (XFG), and employs a skip-gram network to compute the embedding. However, although superficially different, I find that a XFG is very similar to a Program-Dependent-Graph (PDG), which is used in [4] and have been proved to be effective. The two graphs essentially capture very similar information, and it is unclear what's the additional bit that a XFG provides. Skip-gram is also a standard technique to compute the embedding unsupervisedly. Thus, the novelty of this work is thin to me. Second, the evaluation is not convincing enough. The paper claims that the techniques are proposed to compute embeddings for multiple tasks, but I find that the evaluated tasks are a bit artificial. In particular, for the three tasks, (i.e., algorithm classification, device mapping, and thread coarsening factor prediction), it is unclear how much code semantics is needed to accomplished these tasks? These tasks are likely to be determined by looking at the coding patterns, which can be learned from the syntax without understanding the semantics. It could be helpful to evaluate some more fine-grained tasks that are known to leverage semantics information, such as code suggestions, and those evaluated in [4]. Further, the evaluation results for 6.2 and 6.3 are also not convincing that the proposed approach is the best approach in these tasks. In particular, I find that Table 5 is hard to interpret. If the prediction is among {1,2,4,8,16,32}, how is the speedup data computed from the prediction and the ground truth? Last but not least, another recent work in this space is missed: Xu et al., Neural Network-based Graph Embedding for Cross-Platform Binary Code Similarity Detection, CCS 2017